# Fairness-aware Contrastive Learning with Partially Annotated Sensitive Attributes

**Fengda Zhang**[1], **Kun Kuang**[1,2]*, **Long Chen**[3], **Yuxuan Liu**[1], **Chao Wu**[1], **Jun Xiao**[1]
[1]Zhejiang University, [2]Key Laboratory for Corneal Diseases Research of Zhejiang Province
[3]The Hong Kong University of Science and Technology
{fdzhang,kunkuang,yuxuanliu,chao.wu}@zju.edu.cn,
zjuchenlong@gmail.com, junx@cs.zju.edu.cn

## Abstract

Learning high-quality representation is important and essential for visual recognition. Unfortunately, traditional representation learning suffers from fairness issues since the model may learn information of sensitive attributes. Recently, a series of studies have been proposed to improve fairness by explicitly decorrelating target labels and sensitive attributes. Most of these methods, however, rely on the assumption that fully annotated labels on target variable and sensitive attributes are available, which is unrealistic due to the expensive annotation cost. In this paper, we investigate a novel and practical problem of **F**air **U**nsupervised **R**epresentation **L**earning with **P**artially annotated **S**ensitive labels (FURL-PS). FURL-PS has two key challenges: 1) how to make full use of the samples that are not annotated with sensitive attributes; 2) how to eliminate bias in the dataset without target labels. To address these challenges, we propose a general **Fair**ness-aware **C**ontrastive **L**earning (*FairCL*) framework consisting of two stages. Firstly, we generate contrastive sample pairs, which share the same visual information apart from sensitive attributes, for each instance in the original dataset. In this way, we construct a balanced and unbiased dataset. Then, we execute fair contrastive learning by closing the distance between representations of contrastive sample pairs. Besides, we also propose an unsupervised way to balance the utility and fairness of learned representations by feature reweighting. Extensive experimental results illustrate the effectiveness of our method in terms of fairness and utility, even with very limited sensitive attributes and serious data bias.

## 1 Introduction

Learning powerful representation takes an important role in visual recognition, and there are a lot of works proposed to learn visual representations (Bengio et al., 2013; Kolesnikov et al., 2019; Wang et al., 2020a; Liu et al., 2022). Among them, contrastive learning achieves state-of-the-art performance on various vision tasks (Tian et al., 2020; Chuang et al., 2020). Contrastive learning first generates views from original images by random data augmentation, and the views from the same image are defined as positive samples. Then the model can learn effective representations by closing the distance between representations of positive samples, while being protected from mode collapse via an additional module such as negative samples (Chen et al., 2020a; He et al., 2020; Chen et al., 2020b), momentum update (Grill et al., 2020), and stopping gradient (Chen & He, 2021).

Unfortunately, traditional representation learning methods ignore potential fairness issues, which becomes an increasing concern as recognition systems are widely used in the real world (Zemel et al., 2013; Madras et al., 2018; Creager et al., 2019; Lv et al., 2023). For example, the model trained by contrastive learning may learn the information of sensitive attributes (*e.g.,* gender, race) by using it as a shortcut to minimize the distance between representations of positive samples in the training stage, since the positive samples have the same sensitive attributes. As a result, decisions based on biased representation models may discriminate against certain groups or individuals in practice, by using spurious correlations between predictive target and sensitive attributes (Wang

---

*Corresponding author.

et al., 2020b; Park et al., 2021; Zhang et al., 2021). Therefore, how to develop a fair representation model is of paramount importance for both academic research and real applications.

Most of existing works achieve fairness via decorrelating target labels and sensitive attributes explicitly, which rely on the data annotations (Mehrabi et al., 2021; Wu et al., 2022; Zhang et al., 2022; Zhu et al., 2022). However, assuming that all data have fully annotated labels can be unrealistic (Liu et al., 2016; Zhang et al., 2020b; Shao et al., 2021; Jung et al., 2022; Song et al., 2023). In many real scenarios, the target labels and even downstream tasks are not provided, and all we have are images and limited annotations of sensitive attributes. Data labels require additional expensive cost of human annotations, which naturally leads us to ask the following question: *Can we train a fair unsupervised representation model with only partially annotated sensitive attributes?*

In this paper, we investigate a practical and novel problem of Fair Unsupervised Representation Learning with Partially annotated Sensitive attributes (FURL-PS). Our goal is to utilize the images and limited sensitive labels to learn visual representations that can be used for various downstream tasks of visual recognition, while achieving fairness by being minimally correlated with sensitive attributes. It is challenging to solve the proposed problem. Firstly, most samples are not labeled with sensitive attributes. A natural idea is to pseudo-label the unlabeled data by a sensitive attribute classifier. However, it is not advisable to train a representation model on the data with pseudo-sensitive labels, since the noises in pseudo labels may severely affect the fairness performance. Secondly, there may be data imbalance between demographic groups. Assuming that the female group has a large proportion of samples of blond hair, while the male group has the opposite proportion. As a result, the models trained on the above biased data may learn spurious correlation between gender and blond hair. Unfortunately, it is difficult to balance the data distribution of different groups without the prior of downstream tasks or annotated target labels. Generally, FURL-PS problem has two main challenges: 1) How to make full use of the data that are not annotated with sensitive attributes? 2) How to balance the possible agnostic bias in data without target labels?

To address these challenges, our idea is to construct a balanced dataset annotated with sensitive labels based on the original dataset, and then train a representation model with fair contrastive learning on the unbiased dataset. We propose a two-stage Fairness-aware Contrastive Learning (*FairCL*) framework to implement the above idea. In the first stage, we design a semi-supervised learning algorithm to train the image attribute editor with limited sensitive labels, which is used to edit the pre-defined sensitive attributes of a given image. In the second stage, we train a representation model by fair contrastive learning with balanced augmentation. Specifically, based on the image attribute editor, we can generate contrastive sample pairs, which share the same visual information apart from sensitive attributes (*e.g.,* male and female), for each sample in the original dataset. By closing the distance between representations of contrastive sample pairs, the model can learn powerful and fair representations. Our approach has two advantages: 1) we can get the utmost out of unlabeled images by generating samples with given sensitive attributes from them; 2) the augmented dataset is unbiased, since it consists of contrastive sample pairs and thus the data proportions are naturally balanced for different demographic groups. Furthermore, we also develop an unsupervised way to balance the utility and fairness of learned representations by feature reweighting.

We validate the effectiveness of our method on two facial attribute recognition datasets: CelebA (Liu et al., 2018) and UTK-Face (Zhang et al., 2017). Extensive experimental results show that the proposed method outperforms the existing unsupervised learning methods in terms of both classification accuracy and fairness, and even achieves comparable performance with the semi-supervised methods that require annotations on the target labels. Besides, our method is robust to the ratio of sensitive labels and severity of data bias. Furthermore, we also show the extensibility of our general framework to different contrastive learning algorithms through experiments.

**Main Contributions:** 1) To the best our knowledge, we are the first one to propose the practical and challenging problem of Fair Unsupervised Representation Learning with only Partially annotated Sensitive attributes (FURL-PS). 2) We develop the Fairness-aware Contrastive Learning (*FairCL*) framework to solve the proposed problem, which can be compatible with all of contrastive learning algorithms to learn a fair and powerful representation model. 3) Extensive experiments illustrate the effectiveness of our proposed method in terms of fairness and utility.

## 2 RELATED WORK

**Fairness in Unsupervised and Semi-supervised Learning.** There are three branches to guarantee fairness in unsupervised learning. Firstly, fair feature selection, a pre-processing paradigm, finds a subset of features that preserve the original information as much as possible while being minimally correlated with sensitive attributes (Grgic-Hlaca et al., 2016; Grgić-Hlača et al., 2018; Xing et al., 2021). However, this kind of methods is designed for structured data, and cannot be applied to images data and deep models. Secondly, fair clustering balances the distribution of different subgroups formed by sensitive attributes in each cluster, but it cannot yield a model for various downstream tasks (Chierichetti et al., 2017; Kleindessner et al., 2019; Li et al., 2020). At last, some studies based on fair representation learning have been bringing a paradigm for fair unsupervised learning (Louizos et al., 2015; Raff & Sylvester, 2018). As for fair semi-supervised learning, most existing methods first pseudo-label the unlabeled data via a classifier, and then train a model on these data with fairness constraints (Jung et al., 2022; Zhang et al., 2020b;c). However, the pseudo-label noise may exacerbate model unfairness in turn. Instead, our proposed method does not directly use pseudo-labels when training the fair representation model.

**Contrastive Learning.** Self-supervised contrastive learning provides a representation learning paradigm without target labels, and achieves better accuracy than the state-of-the-art methods on various tasks (Xiao et al., 2020; Zhang et al., 2020a). Some methods such as *SimCLR* (Chen et al., 2020a) and *MoCo* (He et al., 2020) first define the positive/negative samples as patches generated from the same/different images via random data augmentation, and then train a representation model by closing/pushing away the distance between representations of positive/negative samples. Recent studies argue that negative samples are not necessary for contrastive learning, and they use some techniques, *e.g.,* momentum update (Grill et al., 2020) and stopping gradient (Chen & He, 2021) to protect the model from mode collapse instead of negative samples. Afterwards, supervised contrastive learning outperforms other state-of-the-art methods based on traditional cross-entropy loss (Khosla et al., 2020). However, existing self-supervised contrastive learning methods ignore potential fairness issues. To this end, *FSCL* proposes a fair supervised contrastive loss to train a fair representation model (Park et al., 2022). However, *FSCL* relies on target labels and sensitive attributes. Besides, *FSCL* is based on supervised contrastive learning which needs negative samples, while our proposed framework is general to be applied to any contrastive learning algorithm to improve fairness.

**Image Generation.** Our proposed method involves the task of image attribute editing, which takes an image as input and aims to generate a new image with desired attributes while preserving other details (Liu et al., 2019; He et al., 2019; Dogan & Keles, 2020). We emphasize that advances in the field of image attribute editing can help improve the performance of our work, since the subsequent methods can also be used here. Some studies aim to construct a balanced and unbiased dataset by data augmentation (Ramaswamy et al., 2021). However, they need the prior of downstream task to generate new samples. Recent works have proposed to evaluate counterfactual fairness by generating counterfactual samples (Denton et al., 2019; Joo & Kärkkäinen, 2020; Dash et al., 2022). Different from them, we consider a more challenging problem to train a fair representation model. Moreover, we consider a more practical setting where there are no target labels and fully annotated sensitive attributes.

## 3 METHOD

In this section, we start with a brief introduction of the problem formulation of FURL-PS and overall flow of our proposed method in Sec. 3.1. Then we display how to generate augmented samples of different sensitive attributes with limited annotated sensitive labels in Sec. 3.2. We elaborate on how to execute fair contrastive learning with balanced augmentation in Sec. 3.3. Lastly, to balance the trade-off between utility and fairness of learned representations, we propose a feature reweighting module for those sensitive attribute-dependent sub-features in Sec. 3.4.

### 3.1 PROBLEM FORMULATION AND OVERALL FLOW

Assume that we have $n$ original images $\{x_k\}_{k=1,2,..,n}$, where $x_k \in \mathcal{X} \subset \mathbb{R}^d$. Labeled dataset is denoted as $D_l = \{x_k, s_k\}_{k=1}^{n_l}$, where $n_l$ is the number of images with annotated sensitive labels, and

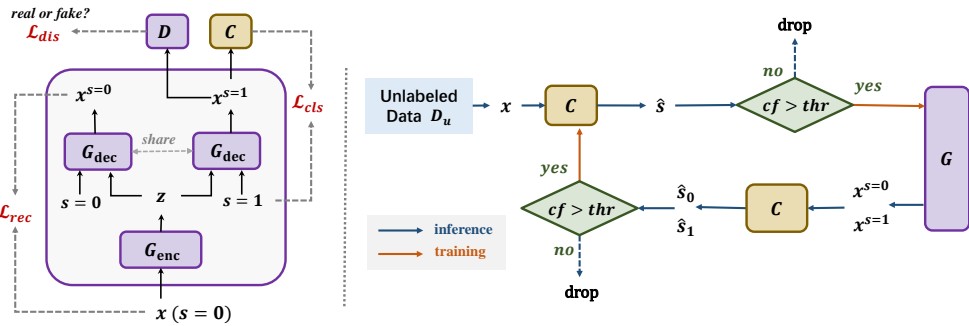

(a) Architecture of Image Attribute Editor $G$      (b) Training Classifier $C$ and Generator $G$ by Mutual Promotion

Figure 1: Semi-supervised Learning of Image Sensitive Attribute Editor.

$s_k \in \{0, 1, ..., M_S - 1\}$ represents the sensitive attribute label (*e.g.,* male and female). Unlabeled dataset is denoted as $D_u = \{x_k\}_{k=n_l+1}^n$. The target labels $\{y_k\}_{k=1,2,..,n}$ are not available in the training state, where $y_k \in \{0, 1, ..., M_Y - 1\}$. In this paper, we assume that both target labels and sensitive attributes are binary variables for convenience, *i.e.,* $M_S = M_Y = 2$. We emphasize, however, that our proposed problem and framework can be easily generalized to multivariate setting. The goal is to train an effective and fair encoder network $F(\cdot)$ that maps the image $x_k \in \mathcal{X}$ into representation $h_k \in \mathcal{H}$, where the representations can be used for various downstream tasks while not discriminating against demographic groups with given sensitive attributes.

Our proposed method consists of two stages: 1) ***Contrastive Sample Generation*** and 2) ***Fairness-aware Contrastive Learning***. We first define the *contrastive samples* as a pair of images that share the same visual information except for sensitive attributes. In the first stage, our goal is to prepare contrastive samples $\{(x_k^0, x_k^1)\}_{k=1,2,..,n}$ based on the original dataset. To achieve it, we design a semi-supervised algorithm to train an image sensitive attribute editor $G(\cdot, \cdot)$, which takes an image $x_k$ and sensitive attribute $s \in \{0, 1\}$ as input and can map the original image to a new image $x_k^s$ with given sensitive attribute $s$ while keeping other information as unchanged as possible. In the second stage, we execute fairness-aware contrastive learning on the augmented dataset to train a representation model $F(\cdot)$ without any target label or sensitive label.

## 3.2 CONTRASTIVE SAMPLE GENERATION WITH LIMITED SENSITIVE ATTRIBUTE LABELS

We start with training a generative model used to generate contrastive samples. We implement it based on AttGAN (He et al., 2019), but we emphasize that any approach designed for image attribute editing can be adapted to be a backbone method. The training architecture of generative model $G$ is shown in Figure 1(a). The encoder $G_{enc}$ maps an input image $x$ to latent representation $z$. Then the decoder $G_{dec}$ takes $z$ and sensitive attributes as input, and generates images with corresponding sensitive attributes. There are three loss optimized jointly: 1) discriminative loss $l_{dis}$ given by the discriminator $D$ guaranteeing that the generated images look realistic enough, 2) classification loss $l_{cls}$ guaranteeing that the generated images have given sensitive attributes, and 3) reconstruction loss $l_{rec}$ given by the classifier $C$ encourages the generator to preserve the sensitive attribute-excluding information as much as possible.

Note that the sensitive labels are needed in the training stage of the image attribute editor $G$, while we only have limited annotated sensitive sensitive labels. To this end, we develop a semi-supervised learning algorithm to train the sensitive attribute classifier and image editor by making them mutually promote each other. We first train a classifier $C$ and image editor $G$ on the labeled data. Then, as shown in Figure 1(b), we assign the predictive sensitive labels to the unlabeled data by $C$. We only select pseudo-labeled samples with confidence $cf$ (*i.e.,* Softmax output score) above a certain threshold $thr$, and train the generative model $G$ on them. Afterwards, we use $G$ to generate additional images with sensitive labels, of which high-confidence images are used to train the classifier $C$. We repeat the above steps until there is no new high-confidence data. Note that the classifier $C$ is used to pseudo-label the unlabeled data, thereby providing the generator $G$ with more annotated training data. Meanwhile, the generator $G$ also generates additional high-quality training data for

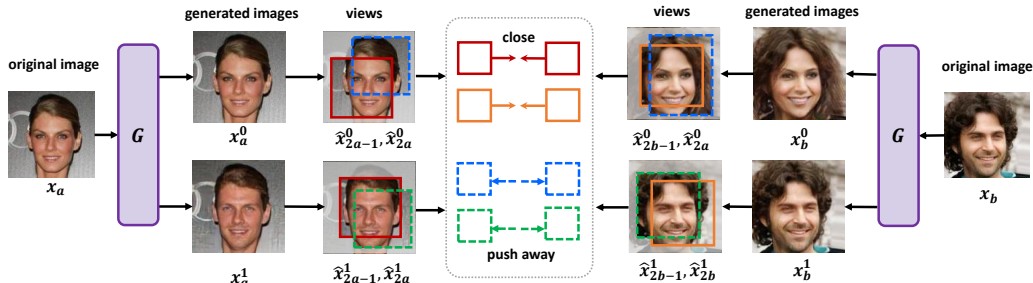

Figure 2: Training Flow of Fairness-aware Contrastive Learning (taking *SimCLR* as an example). Our proposed *FairCL* closes the distance between representations of *positive samples* (*i.e.,* the views from the same original image but have different sensitive attributes), and pushes away the distance between representations of *negative samples* (*i.e.,* the views from different original images but have the same sensitive attributes).

the classifier $C$. In this way, the two can promote each other. Please refer to Appendix A for the details of the above algorithm. The discussion of hyper-parameter $thr$ is in Appendix B.

Here we get an image attribute editor $G$, based on which we can generate an augmented dataset $\{(x_k^0, x_k^1)\}_{k=1,2,..,n}$ from original dataset, even without the knowledge of the sensitive attributes of the original images. It is worth mentioning that we only use pseudo-sensitive labels in the contrastive sample generation stage, which avoids the direct impact of pseudo-label noise on learning fair representation. Furthermore, the augmented dataset is unbiased and balanced for sensitive attributes.

## 3.3 FAIRNESS-AWARE CONTRASTIVE LEARNING WITH BALANCED AUGMENTATION

Based on the balanced augmented dataset generated by the image attribute editor $G$, our goal is to train a fair and powerful representation model. To this end, we develop a fairness-aware contrastive learning framework, and elaborate how it works based on *SimCLR*. However, we emphasize that our proposed general framework is not confined to *SimCLR* but can be easily applied to any contrastive learning algorithm. The key idea is to define the positive samples as the contrastive samples with different sensitive attributes generated by the image attribute editor $G$, and define the negative samples as the views generated from the different images with the same sensitive attributes. Based on it, for a minibatch of contrastive image pairs $\{(x_k^0, x_k^1)\}_{k=1,2,..,N}$, we first generate views $\{(\hat{x}_{2k-1}^0, \hat{x}_{2k}^0, \hat{x}_{2k-1}^1, \hat{x}_{2k}^1)\}_{k=1,2,..,N}$ by data augmentation. An encoder $F(\cdot)$ maps the views into representations $\{(\hat{h}_{2k-1}^0, \hat{h}_{2k}^0, \hat{h}_{2k-1}^1, \hat{h}_{2k}^1)\}_{k=1,2,..,N}$, then a projection head $P(\cdot)$ maps $h_k$ into another representations $\{(\hat{z}_{2k-1}^0, \hat{z}_{2k}^0, \hat{z}_{2k-1}^1, \hat{z}_{2k}^1)\}_{k=1,2,..,N}$ for contrastive learning. Then we define fairness-aware contrastive loss as:

$$L^{Fair} = -\sum_{k=1}^{N} \log \frac{\exp(\hat{z}_{2k-1}^0 \cdot \hat{z}_{2k-1}^1/\tau)}{\sum_{l \neq k} \exp(\hat{z}_{2k}^0 \cdot \hat{z}_{2l}^0/\tau) + \sum_{l \neq k} \exp(\hat{z}_{2k}^1 \cdot \hat{z}_{2l}^1/\tau)}, \quad (1)$$

where $\tau$ is a temperature parameter. As shown in Figure 2, a fair and effective representation model can be trained on the balanced augmented dataset with the proposed fairness-aware contrastive loss.

## 3.4 BALANCE UTILITY AND FAIRNESS VIA FEATURE REWEIGHTING

There is often a trade-off between utility and fairness of representation (Zhao & Gordon, 2019). To balance them without target labels, we propose the feature reweighting module. Our idea is to identify the sensitive attribute-dependent sub-features and reweight them when computing the similarity/distance between representations in contrastive learning. Intuitively, larger weights for those sensitive attribute-dependent sub-features will result in a fairer model.

The challenge is to judge whether a sub-feature is related to a sensitive attribute. Suppose that we train a linear classifier $C_{probe}$ to classify sensitive attributes, which takes the representations generated by the fixed encoder $F(\cdot)$ and projection head $P(\cdot)$ as input. An intuition is that those sensitive attribute-dependent sub-features are easier to activate when predicting sensitive attributes,

and the corresponding parameters of trained classifier $C_{probe}$ will have a larger absolute value. Based on it, we propose a simple but effective solution. We alternately optimize the representation model $F(\cdot)$, $P(\cdot)$ and an additional linear classifier $C_{probe}$. We can strengthen the fairness constraint by assigning more weights to those sub-features where the absolute values of corresponding classifier parameters are large. We use the soft selection which is more flexible and general. Assuming that the parameters of $C_{probe}$ is $(\theta_1, \theta_2, ..., \theta_d)$, where $d$ is the number of dimensions of latent representation. Then we can compute the weights $w = (w_1, w_2, ..., w_d)$ of sub-features as:

$$w_i = \frac{\exp(\alpha \cdot |\theta_i|)}{\sum_{j=1}^{d} \exp(\alpha \cdot |\theta_j|)}, \tag{2}$$

where $\alpha \in \mathbb{R}$ is a scaling parameter, and a larger $\alpha$ means a stronger fairness constraint. Now we can balance the utility and fairness of learned representation in fairness-aware contrastive learning via the weights $w$ as the following:

$$L^{FairW} = -\sum_{k=1}^{N} \log \frac{\exp(\hat{z}_{2k-1}^0 \cdot \hat{z}_{2k-1}^1 \cdot w/\tau)}{\sum_{l \neq k} \exp(\hat{z}_{2k}^0 \cdot \hat{z}_{2l}^0 \cdot w/\tau) + \sum_{l \neq k} \exp(\hat{z}_{2k}^1 \cdot \hat{z}_{2l}^1 \cdot w/\tau)}. \tag{3}$$

## 4 EXPERIMENTS

### 4.1 EXPERIMENTAL SETUP

**Datasets.** We validate our method on the following datasets: 1) **CelebA** (Liu et al., 2018) is a dataset with over 200k facial images, each with 40 binary attributes labels. In this paper, we follow the setting of the previous works (Park et al., 2022). We select *Male* (*m*) and *Young* (*y*) as the sensitive attributes, and set *Attractive* (*a*), *Big Nose* (*b*), and *Bags Under Eyes* (*e*) as target attributes, which have the highest Pearson correlation with the sensitive attributes. Besides, to verify the performance of our method in the setting of multi-target labels and multi-sensitive attributes, we also set {*Male*, *Young*} as sensitive attribute and {*Big Nose*, *Bags Under Eyes*} as target label. The downstream task is unknown and only 5% sensitive attribute labels are available in the training stage. 2) **UTK-Face** (Zhang et al., 2017) contains over 20k facial images, each with attributes labels. We first define a binary sensitive attribute *Young* based on whether age is under 35 or not and construct a task to predict whether the facial image is *Male* or not. We further validate the robustness of our method to the ratio of sensitive labels and data bias on UTK-Face dataset. More details are in Appendix F.

**Evaluation Metrics.** The goal of FURL-PS is to learn a fair and powerful representation model. To validate the fairness and utility of the learned representations, we train a linear classifier on top of the frozen representation model and then use the test performance of the classifier as a proxy for representation quality. In this paper, we use Equal Odds (EO) (Hardt et al., 2016), one of the most commonly used notion of group fairness (Dwork et al., 2012), as the fairness metric:

$$\overline{\sum}_{\forall y, \hat{y}} \left| P_{s^0}(\hat{Y} = \hat{y} \mid Y = y) - P_{s^1}(\hat{Y} = \hat{y} \mid Y = y) \right|, \tag{4}$$

where $\overline{\sum}$ is the averaged sum, $Y$ is target label, $\hat{Y}$ is predictive label given by the classifier, and $s^0, s^1 \in S$ is the value of sensitive attributes. Following (Jung et al., 2022), we extend EO to multi-sensitive attribute setting:

$$\max_{\forall s^i, s^j \in S} \overline{\sum}_{\forall y, \hat{y}} \left| P_{s^i}(\hat{Y} = \hat{y} \mid Y = y) - P_{s^j}(\hat{Y} = \hat{y} \mid Y = y) \right|, \tag{5}$$

where $s^i, s^j \in S$ is the value of sensitive attributes. A smaller EO means a fairer model. Besides, we use top-1 accuracy (%) to measure the effectiveness of learned representations.

**Baselines.** To our best knowledge, there is no existing work focusing on dealing with the problem of FURL-PS. Therefore, we construct some powerful baselines by combining the SOTA methods to solve partially annotated sensitive labels and the advanced methods designed for fair unsupervised representation learning. Specifically, *CGL* (Jung et al., 2022) is the SOTA method to solve the problem of partially annotated sensitive attribute labels by assigning pseudo sensitive labels based on confidence. *VFAE* (Louizos et al., 2015) is a fair representation learning method based on a variational autoencoding architecture with priors that encourage latent factors of variation

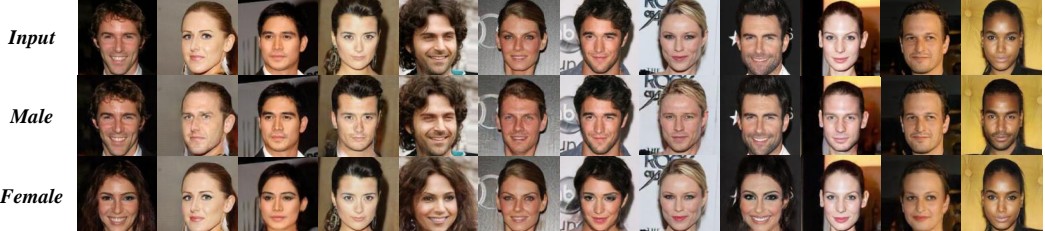

Figure 3: Illustration of contrastive samples generated by sensitive attribute editor.

Table 1: **Unsupervised attribute classification results on CelebA**. To evaluate the quality of representation, we measure equalized odds (EO) and top-1 accuracy (Acc.) of trained linear classifier on CelebA dataset with 5% annotated sensitive attributes. A smaller EO means a fairer model. T and S represent target and sensitive attributes, respectively.

| Method | T=a / S=m | | T=a / S=y | | T=b / S=m | | T=b / S=y | | T=e / S=m | | T=e / S=y | | T=b&e / S=y | | T=a / S=y&m | |
|---|---|---|---|---|---|---|---|---|---|---|---|---|---|---|---|---|
| | EO | Acc. | EO | Acc. | EO | Acc. | EO | Acc. | EO | Acc. | EO | Acc. | EO | Acc. | EO | Acc. |
| CGL+VFAE | 19.1 | 72.7 | 16.2 | 74.0 | 15.5 | 78.3 | 10.6 | 78.8 | 7.6 | 79.6 | 6.9 | 79.5 | 10.2 | 68.7 | 28.7 | 72.7 |
| CGL+GRL | 21.3 | 73.4 | 15.6 | 74.4 | 13.1 | 79.6 | 10.9 | 79.5 | 7.4 | 79.9 | 6.1 | 80.2 | 9.8 | 69.6 | 26.9 | 73.8 |
| SimCLR | 36.2 | 77.1 | 22.5 | 77.1 | 26.9 | 81.5 | 19.2 | 81.5 | 19.7 | 81.2 | 10.6 | 81.2 | 12.8 | 71.5 | 39.6 | 77.7 |
| FairCL (Ours) | **16.8** | 75.3 | **13.1** | 76.9 | **8.4** | 80.0 | **9.2** | 80.3 | **4.2** | 80.8 | **4.5** | 80.5 | **7.8** | 71.3 | **24.5** | 74.1 |

to be independent of sensitive attribute. We implement its unsupervised version and combine it with *CGL* as a baseline (*CGL+VFAE*). We also compare our method with the combination of *CGL* and *GRL* (Raff & Sylvester, 2018), which is an adversarial method used for fair representations (*CGL+GRL*). Since there are few existing methods for fair unsupervised representation learning, we also implement some fair supervised representation learning methods which rely on target labels. Group DRO (*G-DRO*) (Sagawa et al., 2019) is a classical and powerful method for robust and fair learning by learning a set of weights for different data subgroups. *FSCL* (Park et al., 2022) learns fair representations based on supervised contrastive learning. For those 5% samples annotated with sensitive attributes, their target labels are available for *G-DRO* and *FSCL*. The *CGL* strategy is also used for them (*CGL+G-DRO*, *CGL+FSCL*).

## 4.2 CONTRASTIVE SAMPLE GENERATION EXPERIMENTS

We set *Male* as the sensitive attribute, and select some contrastive samples generated by the sensitive attribute editor, which is trained on CelebA dataset with partially annotated sensitive attributes, as shown in Figure 3. As can be seen, the generated contrastive samples remain most of the visual details of the original images, but have different sensitive attributes.

We next discuss the issues of proxy variables. First, for those proxy variables which have the causal/stable correlation with the sensitive attributes, e.g. *Beard* and *Moustache*, we note that the sensitive attribute editor can learn the correlation between them as we expected. For example, the output female does not have a beard, even if the input image is a male face with a beard. Unfortunately, for those proxy variables having the extreme spurious correlation with the sensitive attributes, *e.g., Heavy Makeup* and *Lipstick*, the sensitive attribute editor also learns it. Note that almost all male images have no lipstick (less than 1%), while most female images have lipstick (more than 80%). As a result, as shown in Figure 3, the generated female images sometimes have lipstick, even if the original image is a male face without lipstick. This will result in the representation model not being able to learn information about these proxy variables and thus unable to make accurate predictions about these attributes. We would like to emphasize that this issues is an open problem and, to our best knowledge, there is no existing method that can solve it without any prior. However, we find that our method exhibits robustness to data bias caused by spurious correlations. Specifically, the sensitive attribute editor would not change the attributes which have the high Pearson correlation with the sensitive attributes, *e.g., Big Nose*, due to the reconstruction loss.

Table 2: **Comparison with the methods relying on target labels on CelebA**. 5% samples are annotated with sensitive attributes, and corresponding target labels are available for those target label-dependent methods. Notably, our proposed *FairCL* does not rely on target labels.

| Method | T=$a$ / S=$m$ | | T=$a$ / S=$y$ | | T=$b$ / S=$m$ | | T=$b$ / S=$y$ | | T=$e$ / S=$m$ | | T=$e$ / S=$y$ | | T=$b$&$e$ / S=$y$ | | T=$a$ / S=$y$&$m$ | |
|---|---|---|---|---|---|---|---|---|---|---|---|---|---|---|---|---|
| | EO | Acc. | EO | Acc. | EO | Acc. | EO | Acc. | EO | Acc. | EO | Acc. | EO | Acc. | EO | Acc. |
| *CE* | 27.8 | 77.9 | 19.2 | 77.9 | 20.3 | 82.1 | 14.2 | 82.1 | 16.4 | 82.0 | 11.7 | 82.0 | 9.6 | 71.9 | 36.2 | 77.9 |
| *CGL+G-DRO* | **14.2** | 73.8 | **11.3** | 75.3 | **7.9** | 77.1 | **6.2** | 76.3 | 4.5 | 76.9 | 5.1 | 76.7 | **5.3** | 67.2 | **21.9** | 71.4 |
| *CGL+FSCL* | 17.4 | 75.3 | 13.5 | 76.2 | 9.5 | 79.7 | 9.6 | 79.1 | 5.9 | 81.1 | 5.9 | 80.4 | 8.2 | 69.8 | 25.6 | 74.0 |
| *FairCL* (Ours) | 16.8 | 75.3 | 13.1 | 76.9 | 8.4 | 80.0 | 9.2 | 80.3 | **4.2** | 80.8 | **4.5** | 80.5 | 7.8 | 71.3 | 24.5 | 74.1 |

## 4.3 RESULTS OF FAIRNESS AND ACCURACY ON CELEBA DATASET

We report the classification results of unsupervised methods on CelebA dataset in Table 1. We use equalized odds (EO) and top-1 accuracy (Acc.) of trained linear classifier to evaluate the fairness and utility of learned representations, respectively. A smaller EO means a fairer model. We find that the models trained via *SimCLR* achieves the best accuracy, but suffer from fairness issues. Our proposed *FairCL* based on *SimCLR* improves the fairness of learned representations. *FairCL* outperforms other unsupervised baselines (*CGL+VFAE* and *CGL+GRL*) in terms of EO and accuracy.

We also compare our *FairCL* with the methods relying on target labels, as shown in Table 2. Notably, *FairCL* even outperforms semi-supervised methods in some cases. The reason may be that semi-supervised methods suffers from the issues of pseudo-label noise. Besides, *CGL+G-DRO* achieves excellent EO but has low accuracy due to the over pessimism problem (Hu et al., 2018).

## 4.4 T-SNE VISUALIZATION

To further evaluate the quality of our learned representations and explain how our method works, we sample 1000 images from test dataset and visualize the representations of them via t-SNE (Van der Maaten & Hinton, 2008), as shown in Figure 4. We find that both *FairCL* and *SimCLR* can assign similar features to the images with same target labels. However, *Sim-CLR* also learn information of sensitive attributes, so that the representations given by *SimCLR* can also be divided by the sensitive attributes. In contrast, our proposed *FairCL* focuses on both utility and fairness by closing the distance between representations of contrastive samples, which have similar visual information but have different sensitive attributes.

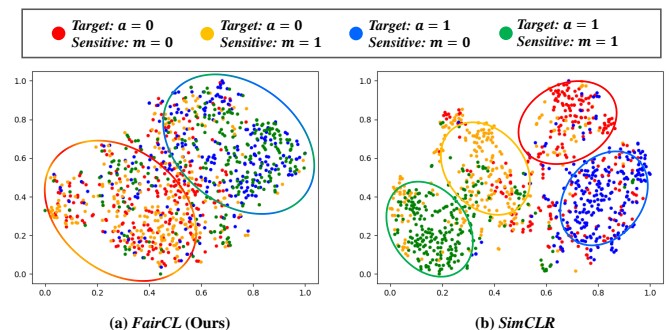

(a) *FairCL* (Ours)   (b) *SimCLR*

Figure 4: t-SNE visualization for the learned representations.

## 4.5 EFFECTIVENESS OF FEATURE REWEIGHTING

We provide an unsupervised way to balance the fairness and utility of learned representation by feature reweighting. To analyze the effectiveness of feature reweighting, we set the scaling parameter $\alpha$ as different values, and train the representation models. The performance of linear classifier in terms of EO and accuracy is reported in Table 3. As can be seen, we find that a larger scaling parameter $\alpha$ can yield a fairer model but with lower accuracy. This is in line with our expectations, since the feature weights will be hard if the scaling parameter $\alpha$ is large, and then the sensitive attribute-dependent features will have a greater impact on the similarity/distance calculation.

## 4.6 COMPATIBILITY WITH CONTRASTIVE LEARNING ALGORITHMS

Our proposed fairness-aware contrastive learning framework is general and flexible that can be used for any contrastive learning algorithm. We have shown the effectiveness of *SimCLR*-based *FairCL* in the previous subsection. To further illustrate the compatibility of *FairCL*, we apply it to *BYOL* (Grill

Table 3: **Effectiveness of feature reweighting on CelebA dataset**. We train the representation models with different scaling parameter $\alpha$ to analyze the effectiveness of feature reweighting.

| $\alpha$ | T=a / S=m | | T=a / S=y | | T=b / S=m | | T=b / S=y | | T=e / S=m | | T=e / S=y | | T=b&e / S=y | | T=a / S=y&m | |
|---|---|---|---|---|---|---|---|---|---|---|---|---|---|---|---|---|
| | EO | Acc. | EO | Acc. | EO | Acc. | EO | Acc. | EO | Acc. | EO | Acc. | EO | Acc. | EO | Acc. |
| $\alpha = 0.5$ | 16.8 | 75.3 | 13.1 | 76.9 | 8.4 | 80.0 | 9.2 | 80.3 | 4.2 | 80.8 | 4.5 | 80.5 | 7.8 | 71.3 | 24.5 | 74.1 |
| $\alpha = 2.0$ | 14.6 | 74.2 | 11.5 | 76.3 | 6.2 | 79.1 | 6.9 | 79.3 | 3.6 | 80.5 | 2.8 | 80.1 | 5.9 | 70.4 | 22.3 | 72.9 |

Table 4: **Compatibility with contrastive learning on CelebA dataset**. We apply our general framework to *BYOL* and denote it as *FairCL\**. The experimental setup is the same as before.

| Method | T=a / S=m | | T=a / S=y | | T=b / S=m | | T=b / S=y | | T=e / S=m | | T=e / S=y | | T=b&e / S=y | | T=a / S=y&m | |
|---|---|---|---|---|---|---|---|---|---|---|---|---|---|---|---|---|
| | EO | Acc. | EO | Acc. | EO | Acc. | EO | Acc. | EO | Acc. | EO | Acc. | EO | Acc. | EO | Acc. |
| *BYOL* | 38.8 | 77.9 | 24.1 | 77.9 | 28.2 | 81.8 | 20.6 | 81.8 | 22.3 | 81.7 | 12.9 | 81.7 | 14.6 | 72.2 | 41.5 | 77.9 |
| *FairCL\** | **19.2** | 76.2 | **15.4** | 77.1 | **13.7** | 80.8 | **11.0** | 80.9 | **7.2** | 81.0 | **6.8** | 80.9 | **9.9** | 71.6 | **28.6** | 74.9 |

et al., 2020), a widely used contrastive learning method without negative samples. Due to the absence of target labels, negative sample-based methods (*e.g., SimCLR*) may incorrectly push the representations of semantically similar samples farther away, which may lead to compromised accuracy on downstream tasks. *BYOL* overcomes this problem by not using negative samples. Therefore, as shown in Table 4, *BYOL* achieves excellent accuracy. However, it also suffers from fairness issues. Notably, *FairCL\** improves EO over it while keeping comparable accuracy.

### 4.7 ROBUSTNESS TO RATIO OF ANNOTATED SENSITIVE LABELS AND DATA BIAS

To further validate the robustness of our proposed method to ratio of annotated sensitive labels and unknown data bias, we run different methods on UTK-Face dataset. We set $Young$ as the sensitive attribute, and the target label is *Male*. Only 5% samples are annotated with sensitive attributes. Besides, the dataset is unbalanced, where the *Young* group has 65% female data and 35% male, while another sensitive group has the opposite gender ratio. As can be seen in Figure 5, our proposed *FairCL/FairCL\** improves fairness compared with *SimCLR/BYOL*, while keeping high accuracy. Our method outperforms other unsupervised methods and even achieves comparable performance with the target label-dependent methods (*CGL+G-DRO* and *CGL+FSCL*), in terms of the trade-off between accuracy and EO, which illustrates that our method is robust to annotation ratio and data bias. More experiments with various settings are in Appendix E.

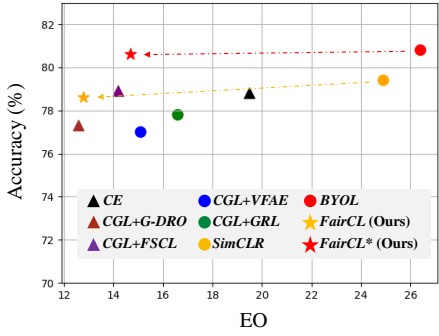

Figure 5: Robustness to ratio of sensitive labels and data bias on UTK-Face. $\triangle$ represents methods relying on target labels.

## 5 CONCLUSIONS

In this paper, we investigate a novel and practical problem of which the goal is to learn a fair and powerful representation model with no target label and limited sensitive attributes. To solve this problem, we develop a general contrastive learning-based framework *FairCL* consisting of two stages: contrastive sample generation and fairness-aware contrastive learning with feature reweighting. Extensive experiments show that our proposed method can yield a fair representation model even with limited sensitive attributes and imbalanced data.

**Limitations and Future Work.** The main limitation is that our work relies on the quality of the generated images. The effectiveness of our method on more challenging domains remains to be explored due to the potential lower-quality generations. We leave it as an open problem for future work. In practice, we suggest the users choose the appropriate generative methods according to the datasets and tasks. Since our proposed fairness-aware framework is compatible with different generative methods, as shown in Appendix E. Moreover, we believe that more effective generative methods can further help improve the quality of representation model trained by our method.

ACKNOWLEDGMENT

This work was supported by the National Key Research & Development Project of China (2021ZD0110400), the National Natural Science Foundation of China (U19B2043, U19B2042, U20A20387, 61976185, 62006207, 62037001), Project by Shanghai AI Laboratory (P22KS00111), Program of Zhejiang Province Science and Technology (2022C01044), the StarryNight Science Fund of Zhejiang University Shanghai Institute for Advanced Study (SN-ZJU-SIAS-0010), Academy Of Social Governance Zhejiang University, and the Fundamental Research Funds for the Central Universities (226-2022-00142, 226-2022-00051).

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

## A  SEMI-SUPERVISED ALGORITHM FOR LEARNING CLASSIFIER AND GENERATOR

The detailed presentation of the method for training generator with limited sensitive attribute labels is shown in Algorithm 1. The semi-supervised algorithm for learning classifier and generator consists of three stages: warm-up training (line 1), dataset extension (lines 2 to 13), and follow-up training (line 14).

---

**Algorithm 1** Semi-supervised Algorithm for Learning Classifier and Generator

---

**Input**: Dataset with sensitive labels $D_l$, unlabeled dataset $D_u$, threshold $thr \in (0, 1)$, initialized classifier $C$, and initialized generator $G$.

1:  Train classifier $C$ and generator $G$ on labeled dataset $D_l$.
2:  Initialize classifier's training set $D_{tr}^C = D_l$, generator's training set $D_{tr}^G = D_l$, and candidate set $D_{cand} = D_u$.
3:  **repeat**
4:      Assign pseudo-sensitive labels to samples in candidate set $D_{cand} = D_u$ using classifier $C$.
5:      Construct a temporary dataset $D_{conf}$ consisting of samples in $D_{cand}$ whose confidences (Softmax scores) are higher than threshold $thr$.
6:      Update (extend) generator's training set $D_{tr}^G = D_{tr}^G \cup D_{conf}$.
7:      Update (reduce) candidate set $D_{cand} = D_{cand} \setminus D_{conf}$.
8:      Train generator $G$ on dataset $D_{tr}^G$.
9:      Edit the sensitive attributes of the samples in dataset $D_{tr}^G$ using generator $G$ to construct a temporary dataset $D_{gen}$.
10:     Construct a temporary dataset $D'_{conf}$ consisting of samples in $D_{gen}$ whose confidences (Softmax scores) are higher than threshold $thr$.
11:     Update classifier's training set $D_{tr}^C = D_{tr}^G \cup D'_{conf}$.
12:     Train classifier $C$ on dataset $D_{tr}^C$.
13: **until** the temporary dataset $D_{conf} = \emptyset$.
14: Train classifier $C$ and generator $G$ on dataset $D_{tr}^C$ and $D_{tr}^G$, respectively.
15: **return** classifier $C$ and generator $G$.

---

## B  DISCUSSIONS OF HYPER-PARAMETERS

### B.1  HYPER-PARAMETER $thr$

**For classifier and generator.** The pseudo-labels given by the classifier may be wrong and not all of the generated samples are high-quality. As a result, there may be some label noises. When we assign a pseudo-label to a sample, intuitively, a high Softmax score means less probability of being classified incorrectly. Therefore, to mitigate the label noises, we introduce the hyper-parameter $thr$ to decide whether an image could be used as a training sample for classifier and generator.

**For representation model.** Before we perform fairness-aware contrastive learning, we first prepare contrastive sample pairs using the trained generator. Then we remove some low-confidence generated samples, since there are a few low-quality generated samples with wrong sensitive attributes, which may introduce bias in the training process of the representation model. Specifically, we compute the average Softmax score for each pair of contrastive samples, and the sample pairs with low average Softmax scores (less than $thr$) will be removed.

**Effect of $thr$.** To explore the effect of hyper-parameter $thr$, we do ablation studies by setting $thr$ to different values. As shown in Table 5, we can observe that the model's Equal Odds (EO) gradually decreases with the increase of $thr$. This experimental phenomenon is consistent with our expectation, since large values of $thr$ guarantee the correctness of the sensitive attributes of the training samples. However, the increase of $thr$ results in a slight decrease in accuracy, which may be due to that the number and diversity of training samples is reduced.

**How to choose $thr$?** There is no criterion to choose $thr$, since the datasets, generative models, training algorithms and requirements are not invariant. However, we can randomly sample from

Table 5: Ablation studies of different values of hyper-parameter $thr$ on CelebA dataset. We set *Male* as sensitive attribute and *Attractive* as target label.

| Method | *FairCL* | | | | *SimCLR* |
|---|---|---|---|---|---|
| $thr$ | 0.99 | 0.90 | 0.80 | 0.00 | / |
| Ratio of removed data (%) | 28.4 | 20.0 | 11.6 | 0.0 | 0.0 |
| Accuracy (%) ($\uparrow$) | 75.0 | 75.3 | 75.8 | 76.7 | 77.1 |
| Equal Odds (EO) ($\downarrow$) | **15.8** | **16.8** | **18.0** | **20.2** | **36.2** |

the data at different $thr$ to assess the quality of the data at different $thr$. In this paper, we set $thr = 0.9$ to balance the fairness and accuracy. In practice, we recommend that the users who are very concerned about fairness set the $thr$ to a large value.

### B.2  Hyper-parameter $\alpha$

We introduce the hyper-parameter $\alpha$ to control the trade-off between accuracy and fairness for our proposed algorithms. As shown in Table 3, a larger scaling parameter $\alpha$ can yield a fairer model but with lower accuracy.

### B.3  Hyper-parameter $\tau$

Hyper-parameter $\tau$ is a temperature parameter, which is widely used in contrastive loss computation for sharper distribution of the Softmax output. In this paper, we follow the setting in (Chen et al., 2020a) and set $\tau = 0.5$.

## C  Experiments of Fairness-accuracy Trade-offs

The classification results of our proposed method and the compared unsupervised baseline methods are reported in Table 1. There are two main metrics we focus on: fairness (EO) and accuracy. From the perspective of multi-objective optimization, we can observe that our *FairCL* strongly Pareto-dominates *CGL+VFAE* and *CGL+GRL*, since *FairCL* outperforms them in terms of both fairness (EO) and accuracy. However, we find that *FairCL* achieves the best performance in terms of fairness, while *SimCLR* achieves the best accuracy. To further compare these two methods in terms of the trade-off between fairness and accuracy, we plot the trade-off curves according to the EO and accuracy during the training process of the linear classifier, as shown in Figure 6. Obviously, our method has a lower EO value when the two achieve the same accuracy, which means that our *FairCL* has a better trade-off between accuracy and fairness.

## D  Beyond Fairness: General Bias Mitigation on Non-facial Dataset

To explore how our proposed framework performs on general bias mitigation, we run different unsupervised algorithms on a non-facial dataset, Dogs and Cats (dog, 2013), which is commonly used in the previous debiasing studies.

**Dataset.** Dogs and Cats dataset consists of dog and cat images. Kim et al. (2019) provide additional annotations for partial images about whether the color of dog/cat is dark or not. We set color as the sensitive attribute, and the task is to predict if the image is a cat or a dog. We construct a biased training set consisting of 2,000 bright cat images, 4,000 bright dog images, 4,000 dark cat images, and 2,000 dark cat images. The Pearson correlation coefficient between target label and sensitive attribute is -0.33, so that the model may learn the spurious correlation between species and colors. We assume that 10% of the samples are annotated with sensitive attributes. The test set is balanced (unbiased) and consists of 2,400 images. We use the same experimental setups (training strategies, models, and hyper-parameters) as before.

**Experimental Results.** We randomly select some generated contrastive sample pairs. as shown in Figure 7. We can observe that the generated contrastive samples remain most of the visual details

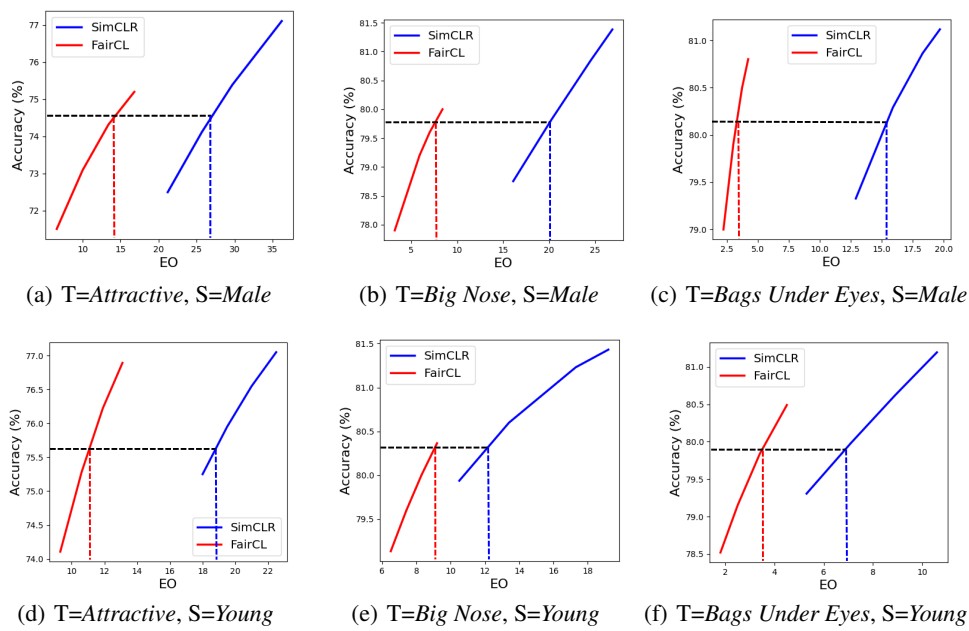

(a) T=*Attractive*, S=*Male*  (b) T=*Big Nose*, S=*Male*  (c) T=*Bags Under Eyes*, S=*Male*

(d) T=*Attractive*, S=*Young*  (e) T=*Big Nose*, S=*Young*  (f) T=*Bags Under Eyes*, S=*Young*

Figure 6: Fairness-accuracy trade-off curves on CelebA dataset. Our *FairCL* achieves a better trade-off between accuracy and fairness.

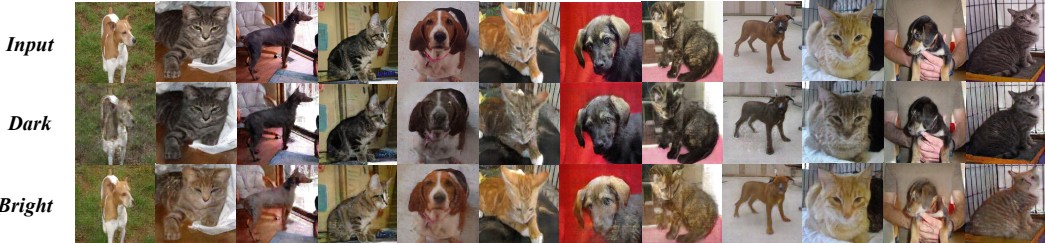

Figure 7: Illustration of contrastive samples generated by sensitive attribute editor.

of the original images, but have different sensitive attributes (colors). We report the classification results in Table 6. Our proposed *FairCL* improves fairness compared to *SimCLR* with only a slight drop in accuracy. Moreover, *FairCL* outperforms all baseline methods in terms of both fairness and accuracy. The above experimental results illustrate that our *FairCL* can prevent representation models from learning spurious correlations.

Table 6: **Unsupervised classification results on Dogs and Cats dataset**. To evaluate the quality of representation, we measure equalized odds (EO) and top-1 accuracy (Acc.) of trained linear classifier on Dogs and Cats dataset with 10% annotated sensitive attributes. A smaller EO means a fairer model. T and S represent target and sensitive attributes, respectively.

| T=*species*, S=*color* | EO (↓) | Acc. (↑) |
|---|---|---|
| *CGL+VFAE* | 10.3 | 85.9 |
| *CGL+GRL* | 9.7 | 86.6 |
| *SimCLR* | 12.4 | 88.1 |
| *FairCL* (Ours) | **8.2** | 87.5 |

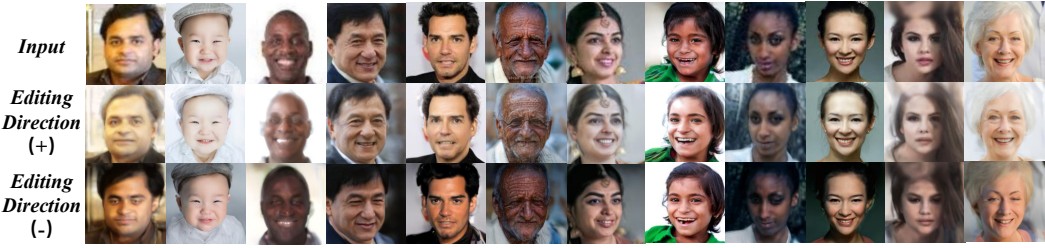

Figure 8: Illustration of contrastive samples generated by sensitive attribute editor on UTK-Face dataset. The generative model is trained by HFGI. (+)/(-) represents that the samples are generated based on the input images plus/minus sensitive attribute-related (race-related) direction.

## E    COMPATIBILITY WITH GENERATIVE FRAMEWORKS

Our proposed fairness-aware contrastive learning is compatible with different generative methods. The descriptions of the contrastive sample generation in the previous section are based on AttGAN (He et al., 2019). To verify the compatibility, we use another generative method, HFGI (Wang et al., 2022), to generate contrastive sample pairs.

**Contrastive Sample Generation based on HFGI.** We design a simple pipeline of contrastive sample generation based on HFGI, which consists of three steps: generator training, classifier training, and image editing. First, we train an inverse GAN using the same experimental settings as HFGI. In this process, we do not need to use any data annotations. Secondly, we train a robust classifier using JTT (Liu et al., 2021) as the training strategy with limited sensitive attributes. The trained classifier is used to assign pseudo-sensitive labels to the unlabeled images. Finally, we get the editing direction w.r.t. sensitive attribute based on InterFaceGAN (Shen et al., 2020). Then we can generate contrastive sample pairs based on the editing direction. We recommend the readers to read the above references for implementation details.

**Experimental Setup.** We use UTK-Face as the dataset. We set race as the sensitive attribute. For convenience, we consider a binary setting, and put white people in one group and other races in another. We would like to state that this grouping is only due to the large proportion of white people in the dataset. The task is to predict the gender for a given image. We construct three unbalanced settings where the Pearson correlation coefficients of race and gender in the training dataset are -0.2, -0.4, and -0.6, respectively. The size of training set is 10,000 in each setting. We also construct a balanced (unbiased) test set with 3,200 images for each setting. We assume that 10% of the samples are annotated with sensitive attributes. To demonstrate the compatibility of our framework to contrastive learning algorithms, we compare *BYOL* and *FairCL** on this dataset, neither of which requires negative samples. Other experimental setups are the same as before.

**Experimental Results.** We randomly select some generated contrastive sample pairs, as shown in Figure 8. We report the classification results in Table 7. We observe that our proposed *FairCL** improves *BYOL* in terms of both fairness and accuracy in three settings, which illustrates that the representation models trained by our framework is more effective and fairer. We also find that the visual quality of the generated images is not as high as we would like, and the racial changes in the edited images are not very obvious. However, the classification results of our method are still greatly improved compared with the baseline, which means that a good representation model can be learned by our proposed framework even the quality of the generated samples is not very high.

**Discussion.** Our proposed framework is based on the generation techniques to prepare contrastive sample pairs for fairness-aware contrastive learning. The above experimental results demonstrate the compatibility of our framework to different generative methods. Moreover, we would like to emphasize that the goal of our study is to learn fair and effective representations, and non-high-quality generative samples do NOT mean bad representations. At last, we believe that the applicability of our method will get stronger and stronger with the development of generation techniques.

Table 7: Unsupervised classification results on UTK-Face dataset based on HFGI generative method. The sensitive attribute is *race*, and the task is to predict the *gender*. We consider three settings with different Pearson correlation coefficients of the training set. The test set is balanced w.r.t. both *race* and *gender*. Our *FairCL\** improves *BYOL* in terms of fairness and accuracy in all settings.

| T=*gender*, S=*race* | Pearson=-0.2 | | Pearson=-0.4 | | Pearson=-0.6 | |
|---|---|---|---|---|---|---|
| Method | EO ($\downarrow$) | Acc. ($\uparrow$) | EO ($\downarrow$) | Acc. ($\uparrow$) | EO ($\downarrow$) | Acc. ($\uparrow$) |
| *BYOL* | 2.1 | 89.4 | 8.7 | 87.5 | 13.1 | 86.4 |
| *FairCL\** (Ours) | **1.7** | **90.6** | **5.6** | **89.6** | **8.9** | **89.7** |
| Improvement | -0.4 | 1.2 | -3.1 | 2.1 | -4.2 | 3.3 |

Table 8: Overview of Experimental Settings. Pearson correlation coefficient measures the linear correlation between target variables and sensitive attributes. For CelebA dataset, we do not use *Wearing Lipstick*, *Heavy Makeup*, *Mustache*, or *No Beard* as the target variables, since their sample sizes of minority groups are less than 50 (<0.25%) in test set.

| Dataset | Generative method | Sensitive attribute | Target variable | Pearson coefficient | Ratio of labeled data |
|---|---|---|---|---|---|
| CelebA | AttGAN | *Male* | *Attractive* | -0.40 | 5% |
| | | | *Big Nose* | 0.37 | |
| | | | *Bags Under Eyes* | 0.30 | |
| | | *Young* | *Attractive* | 0.39 | |
| | | | *Big Nose* | -0.29 | |
| | | | *Bags Under Eyes* | -0.23 | |
| CelebA | \ | *Male* | *Wearing Lipstick* | -0.79 | \ |
| | | | *Heavy Makeup* | -0.67 | |
| | | | *Mustache* | 0.25 | |
| | | | *No Beard* | -0.52 | |
| | | *Young* | *Wearing Lipstick* | 0.25 | |
| | | | *Heavy Makeup* | 0.25 | |
| | | | *Mustache* | -0.14 | |
| | | | *No Beard* | 0.12 | |
| UTK-Face | AttGAN | *Age* | *Gender* | -0.30 | 5% |
| | HFGI | *Race* | *Gender* | -0.20 | 10% |
| | | | | -0.40 | |
| | | | | -0.60 | |
| Dogs and Cats | AttGAN | *Color* | *Species* | -0.33 | 10% |

# F    DETAILS OF EXPERIMENTAL SETUP

**Overview of Experimental Setup.** The overview of our experimental settings is shown in Table 8. For CelebA dataset, the sample sizes of *Wearing Lipstick, Male*, *Heavy Makeup, Male*, *Mustache, Female*, and *No Beard, Female* in test set are 47, 22, 0, and 10, respectively. All of them have a fraction less than 0.25% in the test set, so we do not use these attributes in experiments.

**Implementation Details.** We resize the images of CelebA and UTK-Face to 128×128, and use a 5-layer CNN (Krizhevsky et al., 2017) as the encoder of generative model. Besides, the decoder also has 5 layers. Since the quality of some generated samples may not be good enough, we select images based on confidence. Specifically, we use the trained classifier to predict the sensitive attributes of the generated images, and then remove some low-confidence samples whose Softmax output scores are lower than $thr = 0.9$. Besides, to further improve the quality of training data, we use high-confidence of original images instead of corresponding generated samples. We use the same random data augmentation strategies as (Chen et al., 2020a). The projection head $P(\cdot)$ is only used in contrastive learning, and then we remove it. We use the ResNet-18 (He et al., 2016) as encoder model and a MLP as projection head, and train them via weighted fairness-aware contrastive loss for 100 epochs. Afterwards, we train a linear classifier on top of the frozen representation given by encoder $F(\cdot)$ for 10 epochs on the training dataset.

