# OpenReview forum: "Fairness-aware Contrastive Learning with Partially Annotated Sensitive Attributes"
_ICLR.cc/2023/Conference — ICLR 2023 poster_

### Official Review · Reviewer_G4GB · 2022-10-21

**Confidence:** 4
**Clarity, Quality, Novelty And Reproducibility:** The proposed idea is sound, the paper…
**Correctness:** 4
**Technical Novelty And Significance:** 3
**Empirical Novelty And Significance:** 3
**Recommendation:** 5

**Strength And Weaknesses:**

Advantages:
- It is worthwhile to investigate the problem of fair unsupervised representation learning with only partially annotated sensitive attributes.
- The paper is well written.
- The proposed FairCL framework is generally sound, and experimental analysis could validate the framework's effectiveness to some extent.

Drawbacks:
- My main concern is that the proposed method is limited to face recognition-based applications. It is now possible to generate contrastive samples for face images thanks to the advancement of GAN-based techniques. However, doing so for other applications is challenging. Without high-quality augmentations, it is doubtful that this method can be applied in other applications.
- Another major concern is that the authors only show one point on the fairness-accuracy trade-off curve. The majority of existing literature assesses the effectiveness of fairness mitigation algorithms using a fairness utility curve to determine which algorithm achieves the best fairness accuracy trade off. It would be difficult to determine which of the two is superior if only one point on the curve was considered: 1) EO: 39.6, and ACC: 77.7, and 2) EO: 24.5, and ACC: 74.1.
- The evaluation is limited to the two datasets of the face recognition tasks.
- An important baseline is missing. That is, we first train the sensitive attribute classification model using the partial sensitive attribute annotations. Following that, this model can be used to generate pseudo sensitive labels for other unlabelled samples. This method may not work well with difficult real-world datasets. However, for the two datasets used in this study, CelebA and UTK-Face, even a low ratio of labeled sensitive labels (e.g., less than 5%) are sufficient to train high-quality sensitive attribute classification models. The main reason is that the sensitive attribute annotation (male and female) task is much easier than the underlying task, such as Attractive.


**Summary Of The Paper:**

Traditional representation learning methods (for example, contrastive learning), while making significant progress, have potential fairness issues. This paper proposes training the image attribute editor to generate contrastive sample pairs for each sample in the original dataset, which share the same visual information except for sensitive attributes (e.g., male and female). To enable the model to learn powerful and fair representations, it reduces the distance between representations of contrastive sample pairs. The key idea is to first build the debiased training set and then learn the debiased representation. The proposed method outperforms existing unsupervised learning methods in terms of classification accuracy and fairness, according to experimental results on the CelebA and UTK-Face datasets.



**Summary Of The Review:**

Although this work has some merits, I am concerned about the experimental evaluations and the proposed method's real-world applicability.

---

> ### Author Response · Authors · 2022-11-18
> **Response to Reviewer G4GB (1/2)**
>
> We would like to thank you for taking the time to review our paper and for providing detailed and helpful comments! We are particularly pleased that you think the paper is well written, the proposed framework is generally sound, and the proposed problem is worthwhile to investigate.
>
> Below are responses to the concerns raised by you. Please let us know if you require any further information, or if anything is unclear.
>
> ## Q1: Generalization to non-facial dataset and dependencies on high-quality augmentations.
> >My main concern is that the proposed method is limited to face recognition-based applications. It is now possible to generate contrastive samples for face images thanks to the advancement of GAN-based techniques. However, doing so for other applications is challenging. Without high-quality augmentations, it is doubtful that this method can be applied in other applications.
> >The evaluation is limited to the two datasets of the face recognition tasks.
>
> **A1:** Thanks for your suggestions. We have added extra expermiments on non-facial dataset in Appendix D. We also added more experiment with different generative mothods, for example HFGI [1] in Appendix E. We hope the responses in the following can help to address your concerns.
>
> - **Extra experiments of generalization to the non-facial dataset.**
>
>   - **Experimental setup.** We used Dogs and Cats dataset [2], which is commonly used in the previous debiasing studies. We set color as the sensitive attribute, and the task is to predict if the image is a cat or a dog. We constructed a biased training set where the Pearson correlation coefficient is -0.33, so that the model may learn the spurious correlation between species and colors.
>
>   - **Experimental results.** The classification results are shown in the table as below. Our proposed *FairCL* improves fairness compared to *SimCLR* with only a slight drop in accuracy. Moreover, *FairCL* outperforms all baseline methods in terms of both fairness and accuracy. The above experimental results illustrate that our proposed framework can perform well on the non-facial dataset, and can prevent representation models from learning spurious correlations.
>
> | Dogs and Cats dataset, T=*species*, S=*color* |  EO  | Acc. (%) |
> |:----------------------:|:----:|:--------:|
> |       *CGL+VFAE*       | 10.3 |   85.9   |
> |        *CGL+GRL*       |  9.7 |   86.6   |
> |        *SimCLR*        | 12.4 |   88.1   |
> |     *FairCL* (Ours)    |  **8.2** |   87.5   |
>
> - **Extra experiments of generalization to different generative methods.**
>
>   - **Experimental Setup.** We used another generative method, HFGI [1], to generate contrastive sample pairs. We used UTK-Face as the dataset and set race as the sensitive attribute. The task is to predict the gender. We constructed three unbalanced settings where the Pearson correlation coefficients in the training dataset are -0.2, -0.4, and -0.6, respectively. We constructed a balanced (unbiased) test set.
>
>   - **Experimental Results.** We report the classification results in the table as below. Our proposed _FairCL*_ improves *BYOL* in terms of both fairness and accuracy in three settings, which illustrates that the representation models trained by our framework is more effective and fairer. The results also show that our proposed fairness-aware framework is compatible with different generative methods. This means that we can choose the appropriate generation method according to the dataset and task.
>
> |      Pearson correlation coefficient     |   -0.2  |   -0.2   |   -0.4  |   -0.4   |   -0.6  |   -0.6   |
> |:----------------:|:-------:|:--------:|:-------:|:--------:|:-------:|:--------:|
> |      UTK-Face dataset, T=*gender*, S=*race*      |    EO   | Acc. (%) |    EO   | Acc. (%) |    EO   | Acc. (%) |
> |      *BYOL*      |   2.1   |   89.4   |   8.7   |   87.5   |   13.1  |   86.4   |
> | _FairCL* (Ours)_ | **1.7** | **90.6** | **5.6** | **89.6** | **8.9** | **89.7** |
> |    Improvement   |   -0.4  |    1.2   |   -3.1  |    2.1   |   -4.2  |    3.3   |
>
> - **Discussion.**
>
> **It is possible to learn a good representation model even if the generated samples are not of very high quality.** For example, as shown in Figure 8, the visual quality of the generated images is not as high as we would like, and the racial changes in the edited images are not very obvious. However, the classification results of our method are still greatly improved compared with the baseline. Moreover, our general framework will **benefit from the development of generative techniques**. We believe its applicability will become stronger and stronger.
>
> **Corresponding Manuscript Revision:** We have added additional experiments on a non-facial dataset in Appendix D. We also added experiments based on a new generative method (HFGI) in Appendix E.
>
> [1] Tengfei Wang, Yong Zhang, Yanbo Fan, Jue Wang, and Qifeng Chen. "High-fidelity gan inversion for image attribute editing."  CVPR 2022.
>
> [2] Dogs vs. cats. In Kaggle, 2013.

---

> > ### Author Response · Authors · 2022-11-18
> > **Response to Reviewer G4GB (2/2)**
> >
> > ## Q2: Fairness-accuracy trade-off curve.
> > >Another major concern is that the authors only show one point on the fairness-accuracy trade-off curve. The majority of existing literature assesses the effectiveness of fairness mitigation algorithms using a fairness utility curve to determine which algorithm achieves the best fairness accuracy trade off. It would be difficult to determine which of the two is superior if only one point on the curve was considered: 1) EO: 39.6, and ACC: 77.7, and 2) EO: 24.5, and ACC: 74.1.
> >
> > **A2:** We have added experiments to compare our FairCL with baseline by plotting the fairness-accuracy curves in Appendix C.
> >
> > There are two main metrics we focus on: fairness (EO) and accuracy. From the perspective of multi-objective optimization, based on the classification results in Table 1, we can observe that our *FairCL* strongly Pareto-dominates *CGL+VFAE* and *CGL+GRL*, since *FairCL* outperforms them in terms of both fairness (EO) and accuracy. However, we find that *FairCL* achieves the best performance in terms of fairness, while *SimCLR* achieves the best accuracy. To further compare these two methods in terms of the trade-off between fairness and accuracy, we plot the trade-off curves according to the EO and accuracy during the training process of the linear classifier, as shown in Figure 6. Obviously, our method has a lower EO value when the two achieve the same accuracy, which means that __our *FairCL* has a better trade-off between accuracy and fairness.__
> >
> > **Corresponding Manuscript Revision:** We have added additional experiments to evaluate the fairness-accuracy trade-off by plotting trade-off curves in Appendix C.
> >
> > ## Q3: Missing baseline.
> > >An important baseline is missing. That is, we first train the sensitive attribute classification model using the partial sensitive attribute annotations. Following that, this model can be used to generate pseudo sensitive labels for other unlabelled samples. This method may not work well with difficult real-world datasets. However, for the two datasets used in this study, CelebA and UTK-Face, even a low ratio of labeled sensitive labels (e.g., less than 5%) are sufficient to train high-quality sensitive attribute classification models. The main reason is that the sensitive attribute annotation (male and female) task is much easier than the underlying task, such as Attractive.
> >
> > **A3:** Thanks for your suggestion! Please note that CGL [3] is the SOTA method to assign pseudo sensitive labels for unlabeled samples. We constructed some powerful baselines by combining the fairness-aware method and CGL, and we have compared our method with these baselines. We think the compared baseline methods are powerful enough. However, we will add further experiments if you think it is necessary.
> >
> > [3] Sangwon Jung, Sanghyuk Chun, and Taesup Moon. "Learning fair classifiers with partially annotated group labels." CVPR 2022.

---

### Official Review · Reviewer_Vv8C · 2022-10-23

**Confidence:** 5
**Correctness:** 3
**Technical Novelty And Significance:** 3
**Empirical Novelty And Significance:** 3
**Recommendation:** 8

**Clarity, Quality, Novelty And Reproducibility:**

The article is well written and easy to understand. Author proposed a new problem of learning fair representation. They clearly pointed out the obstacle of acquiring the sensitive labels and refer to the idea of contrastive learning, in which generated image as positive samples to solve this problem. The various sensitive attributes in these generated samples ensure the model to implement fair representation learning, which is very novel and effective idea. Though with a good motivation and framework, the comprehensiveness of experiments are barely satisfactory and related explanations are lack of further analysis detailed, especially some hyper-parameters and the pseudo-code.

**Strength And Weaknesses:**

Strength:
1)The first strength of this article is the proposed new problem: FURL-PS, which aims to settle the key issue that sensitive attribute label is high cost to obtain and even suffer privacy policy. It has well practicability and worth to generalize.
2)Besides, a major strength in this method is to implement the dataset adjustment to adapt the contrastive learning in self-supervised manner, thereby design a general framework FairCL. The design solution for the absence of sensitive label is also technically feasible to generate some positive samples that support the fair representations learning. In this case, after identifying the qualities of representations, it makes sense to implement this method and realize the process of fair learning.
3)The corresponding experiments in two datasets are enough to validate its effectiveness and generalizability.

Weakness:
According to my understanding, the performance of this method relies heavily on the quality of generated images as positive sample, which is implemented by off-the-shelf framework AttGAN. In order to verify the generalization of your method, I think you should further discuss the effect of different frameworks of image generation. If possible, you could also provide some failure cases to observe the stability of your model.

**Summary Of The Paper:**

In this manuscript, author attempt to employ the supervision of limited sensitive labels in images for learning fair representations. To address this new task, they refer to the idea of contrastive learning and propose a novel fairness-aware framework. Specifically, to ensure this fairness in contrastive learning, author adopt a generator to obtain some positive samples with various sensitive attributes, which could avoid the effect of sensitive attributes to ensure impartial learning process.

**Summary Of The Review:**

The highlight of this paper falls on their motivation and solution, which has well practicability and worth to generalize. It provides a new direction to implement unbiased and fair representation learning under the limited annotated information.Besides, although the number of experiments is enough, the related analysis is barely satisfactory. Some experiments result still need to further analysis.

---

> ### Author Response · Authors · 2022-11-18
> **Response to Reviewer Vv8C**
>
> Thank you for reviewing our paper. we appreciate it. Thank you for recommending that our paper be accepted. We are happy that you think that our problem and method are novel, and the framework is general. In addition, we are pleased that you appreciate our experiments.
>
> Below are responses to the concerns raised by you. Please let us know if you require any further information, or if anything is unclear.
>
> ## Q1: Generalization to different generative methods.
> >According to my understanding, the performance of this method relies heavily on the quality of generated images as positive sample, which is implemented by off-the-shelf framework AttGAN. In order to verify the generalization of your method, I think you should further discuss the effect of different frameworks of image generation.
>
> **A1:** Thanks for this suggestion, and we think it is helpful to improve the quality of our paper. We have added experiments and discussion based on a new generated method HFGI [1] in Appendix E.
>
> - Experimental Setup. We used UTK-Face as the dataset and set race as the sensitive attribute. The task is to predict the gender. We constructed three unbalanced settings where the Pearson correlation coefficients of *race* and *gender* in the training dataset are -0.2, -0.4, and -0.6, respectively. We also constructed a balanced (unbiased) test set. We used HFGI to generate contrastive sample pairs.
>
> - Experimental Results. We report the classification results in the table as below. We observe that our proposed _FairCL*_ improves *BYOL* in terms of both fairness and accuracy in three settings, which illustrates that the representation models trained by our framework is more effective and fairer. The results also show that our proposed fairness-aware framework is compatible with different generative methods.
>
> |      Pearson correlation coefficient     |   -0.2  |   -0.2   |   -0.4  |   -0.4   |   -0.6  |   -0.6   |
> |:----------------:|:-------:|:--------:|:-------:|:--------:|:-------:|:--------:|
> |      UTK-Face dataset, T=*gender*, S=*race*      |    EO   | Acc. (%) |    EO   | Acc. (%) |    EO   | Acc. (%) |
> |      *BYOL*      |   2.1   |   89.4   |   8.7   |   87.5   |   13.1  |   86.4   |
> | _FairCL* (Ours)_ | **1.7** | **90.6** | **5.6** | **89.6** | **8.9** | **89.7** |
> |    Improvement   |   -0.4  |    1.2   |   -3.1  |    2.1   |   -4.2  |    3.3   |
>
>
> Please refer to Appendix E for more details of experiments.
>
> **Corresponding Manuscript Revision:** We have added experiements based on a new generative method HFGI in Appendix E.
>
> [1] Tengfei Wang, Yong Zhang, Yanbo Fan, Jue Wang, and Qifeng Chen. "High-fidelity gan inversion for image attribute editing."  CVPR 2022.
>
> ## Q2: Failure cases.
> >If possible, you could also provide some failure cases to observe the stability of your model.
>
> **A2:** Our model may be not work well when there is an extremely suprious correlation between unknown target variables and sensitive attributes. For example, in CelebA dataset, almost all male images have no lipstick (less than 1%), while most female images have lipstick (more than 80%). As shown in Figure 3, the generated female images sometimes have lipstick, even if the original image is a male face without lipstick. This will result in the representation model not being able to learn information about these proxy variables and thus unable to make accurate predictions about these attributes. However, we believe the development of robust classification algorithms may help address this issue to make our method more practical.
>
> **Corresponding Manuscript Revision:** We have added more discussion about failure cases in Section 4.2.
>
> ## Q3: Ablation studies on hyperparameters and the pseudo-code.
> >Though with a good motivation and framework, the comprehensiveness of experiments are barely satisfactory and related explanations are lack of further analysis detailed, especially some hyperparameters and the pseudo-code.
>
> **A3:** Thanks for your suggestion! We have provided the pseudo-code of the proposed semi-supervised algorithm in Appendix A. We also addded ablation studies about hyperparameters ($thr$, $\alpha$, $\tau$) in Appendix B.
>
> **Corresponding Manuscript Revision:** We have provided the pseudo-code of the proposed semi-supervised algorithm in Appendix A, and addded experiments and discussion about hyperparameters ($thr$, $\alpha$, $\tau$) in Appendix B.

---

> > ### Comment · Reviewer_Vv8C · 2022-12-08
> > **Concerns Addressed**
> >
> > I would like to thank the authors for their responses and clarifications, which have addressed all my concerns. Particularly, the addtional experimental results have shown the good generalization and compatiblily of the proposed framework.
> > After reading the revision and other reviewers' comments, I insist on keeping my positive score and recommend this submission in view of its strong motivation and novel solution.

---

### Official Review · Reviewer_MhQc · 2022-10-23

**Confidence:** 4
**Correctness:** 4
**Technical Novelty And Significance:** 3
**Empirical Novelty And Significance:** 3
**Recommendation:** 8

**Clarity, Quality, Novelty And Reproducibility:**

The paper was easy to follow and clearly written, backed by appropriate model comparison tables, graph, and model architecture diagrams along with appropriate equations. The proposed method is novel as described above.

**Strength And Weaknesses:**

Dealing with a dataset partially annotated with sensitive labels is a known and common problem in the real world. However, as far as I know, this paper seems to be the first to give the problem a formal name FURL-PS to be able to refer to it with an ease. In addition, the choice of using AttGAN for the generative model is clever because AttGAN was already demonstrated in the original paper with the same CelebA dataset to modify only the intended face attribute. So, in this case, just a sensitive face attribute can be modified according to the already approved paper's demonstration. This paper demonstrated the effectiveness and limitation of using negative samples generated by an image editor with two different versions of the solutions FairCL and FairCL*. It seems like an extra mile the authors took that made the paper's findings even more interesting and useful for the reader. The theory used to balance between utility and the fairness using feature re-weighting in the absence of the target labels, is also simple and clever. The experiments themselves seem to be a high quality in that the authors chose downstream task face attributes that have high Pearson correlation with the sensitive attributes and in that the authors honestly wrote the limitation of the proposed solution in the presence of the "extreme spurious correlation" like in the case of Heavy Makeup and Lipstick. Figure 5 is also very helpful to better understand the effectiveness of the proposed model compared to other prior arts.

Nevertheless, there were some lingering questions/doubts about the approach described. The process of training the image editor G and the sensitive label classifier C with unlabeled data feels like a self-feeding/self-reinforcing loop that can introduce some bias. There wasn't much discussion about this to satisfy my doubts about the proposed model's fairness limitation that may come from this self-feeding training loop. It would have been also interesting if the author listed the Pearson correlation numbers of the face attributes that the author defined as "extremely spurious correlation" vs "spurious correlation" in order to translate the potentially qualitative and intuitive description into a more quantitative description.

**Summary Of The Paper:**

The author aims to bring attention to a known problem by giving it a formal name FURL-PS and definition, and proposes a practical solution, FairCL/FairCL* which is a novel combination of known theory and algorithms previously not having been applied to this particular problem of ML fairness. In particular, the paper demonstrates how to tackle the problem of training a face image encoder with a good ratio between the equalized odds and the face attribute classification accuracy, given a dataset partially annotated with sensitive labels, such as gender and age, and without the use of the target labels of the downstream tasks (in this case, face attribute classification). The paper is backed with appropriate model performance comparison tables and graph to show the effectiveness of the proposed solution FairCL/FairCL* along with some of its limitations.

**Summary Of The Review:**

I recommend to accept this paper. Despite some lingering doubts mentioned before, the theory seems sound, and the empirical results seem comprehensive and promising. The paper is coherently written with clear language backed by appropriate tables and figures to assist easier reading and understanding. The proposed method is novel and tackles an important and practical ML Fairness problem faced in the real world.

---

> ### Author Response · Authors · 2022-11-18
> **Response to Reviewer MhQc**
>
> Thanks a lot for your time and effort in reviewing our paper. We appreciate it! In addition, we are pleased that you appreciate the clarity of the paper, the novelty of our method, and the effort put into the design of our experiments, which you have reflected by your current recommendation.
>
> Below are responses to the concerns raised by you. Please let us know if you require any further information, or if anything is unclear.
>
> ## Q1: Discussion about the potential bias in self-feeding loop.
>
> >The process of training the image editor G and the sensitive label classifier C with unlabeled data feels like a self-feeding/self-reinforcing loop that can introduce some bias. There wasn't much discussion about this to satisfy my doubts about the proposed model's fairness limitation that may come from this self-feeding training loop.
>
> **A1:** Thanks for this question. We added more discussion about the potential bias in the self-feeding loop in Appendix B in the revision.
>
> - **Threshold mechanism for bias mitigation.** Some bias may be introduced in this loop, since the pseudo-labels given by the classifier may be wrong and not all of the generated samples are high-quality. That is why we introduce a threshold mechanism (with a hyperparameter $thr$) to filter training samples for classifier and generator.
>
> - **Extra experiments of the threshold mechanism.** Empirically, we recorded the accuracy of the classifier with and without the threshold mechanism. Compared with the situation without the threshold mechanism, the accuracy of the classifier has been improved after the threshold mechanism is introduced (from 87% to 97%, CelebA dataset, S=*Young*, $thr=0.9$). To further explore the effect of the threshold mechanism, we did ablation studies by setting $thr$ to different values. We can observe that the model's Equal Odds (EO) gradually decreases with the increase of $thr$. This experimental phenomenon is consistent with our expectation, since large values of $thr$ guarantee the correctness of the sensitive attributes of the training samples. The above results further show that the threshold mechanism can effectively alleviate the bias in loop.
> More details about discussion and experiments are in Appendix B.
>
> **Corresponding Manuscript Revision:** We have added more discussion and experiments in Appendix B.
>
> ## Q2: List the Pearson correlation numbers.
> >It would have been also interesting if the author listed the Pearson correlation numbers of the face attributes that the author defined as "extremely spurious correlation" vs "spurious correlation" in order to translate the potentially qualitative and intuitive description into a more quantitative description.
>
> **A2:** Thanks for your suggestions. We have added a table of the overview of experimental settings in Appendix F. The table is detailed and contains Pearson correlation coefficients of different attributes.
>
> Taking the attribute Wearing Lipstick in CelebA dataset as an example, the Pearson correlation coefficient between it and gender is -0.79 (defined as "extremely spurious correlation"). Specifically, few male images have lipstick (less than 1%), while most female images have lipstick (more than 80%). However, the samples of {*Male*, *Wearing Lipstick*} have a fraction less than 0.25% in the test set, so we do not use the attribute *Wearing Lipstick* in experiments. The Pearson correlation coefficient of the attribute *Attractive* and *gender* is -0.40, which is the highest except those unavailable attributes.
>
> Please refer to section 4.2 and Appendix F, if you are interested in more details.
>
> **Corresponding Manuscript Revision:** We have added a table of the overview of experimental settings (including Pearson correlation numbers) in Appendix F. We also improved the presentation in Section 4.2 by providing more quantitative description.

---

### Official Review · Reviewer_kKgZ · 2022-10-24

**Confidence:** 5
**Correctness:** 3
**Technical Novelty And Significance:** 2
**Empirical Novelty And Significance:** 2
**Recommendation:** 6

**Clarity, Quality, Novelty And Reproducibility:**

- Exact details are not clearly presented and it is not clear how to reproduce the result.
   - Line 176: how is *thr* chosen?
   - Line 268: "remove some low-confidence samples" --> this is not rigorous.
   - In fact, the whole paragraph line 171~182 is not very rigorous. It is not clear how to reproduce based on the description. Also, it is not clear whether the iterative steps described in the paragraph would always work.

- The components of the method are careful combination of existing techniques rather than novel development of new schemes. The setting seems to be new, but it is not clear whether the proposed method is generalizable to other datasets other than face-related image datasets.

**Strength And Weaknesses:**

Strength:
- As mentioned above, the method combines multiple methods to achieve the goal, which is to learn fair representation with partially annotated sensitive label and without target label. It seems to be the first work to consider this specific setting.
- The generative model example and t-sne visualization provides necessary qualitative results.
- The proposed method is compatible to several contrastive learning methods.
- The method can generate similar quality of fair representation compared to the methods with target labels.

Weakness:
- There are many hyperparameters (*thr* for generative model, $\alpha$, $\tau$), and the effect of them is not considered or analyzed. There is only one table on $\alpha$. Particularly, I think the effect of *thr* should be very important since the generative models are typically sensitive to the hyperparameter choice.
- The method is highly dependent on the GAN-based generative model, and when the dataset is not face-related images like CelebA or UTKFace, it is not clear how the method would perform.


**Summary Of The Paper:**

The paper proposes a variation of contrastive learning that learns fair representation with partially annotated sensitive attribute labels. It combines several components together to achieve the goal: generative model that can generate "counterfactual" examples with different sensitive labels for making the positive and negative samples for contrastive learning, pseudo-label prediction model to estimate the sensitive attributes, and feature re-weighting scheme for balancing utility and fairness. The resulting method is shown to achieve positive results on benchmark dataset for (unsupervised) learning fair representations with partially annotated sensitive labels.

**Summary Of The Review:**

As mentioned above, while the problem setting seems to be new (but, it is also inspired by recent work (Jung et al, 2022) and is not very hard to think of) and there are some positive aspects in the paper, I think the overall method simply combines existing methods (which undermines the novelty) and the description of the method is not rigorous/general enough for a publication at ICLR.

**** Post-rebuttal
I think some limitations still exists since the proposed method looks highly dependent on the examined dataset regarding face images, for which synthetic data is relatively easy to generate via GAN. But, the proposed problem setting indeed can be regarded as novel and the paper also has some value of tacking it for the first time. The experimental results for the considered dataset is promising, too, so I raised my score to 6. I think the authors need to at least describe the limitation of their method as well.

---

> ### Author Response · Authors · 2022-11-18
> **Response to Reviewer kKgZ (1/3)**
>
> We would like to thank you for providing detailed and helpful comments! We are particularly pleased that you point that the paper is the first work to consider this specific setting (FURL-PS). In addition, thank you for noting that our unsupervised framework can generate similar quality of fair representation compared to the methods with target labels and is compatible to several contrastive learning methods.
>
> Below are responses to your concerns. Please let us know if you require any further information, or if anything is unclear.
>
> ## Q1: More ablation studies on different hyperparameters.
> >There are many hyperparameters ($thr$ for generative model, $\alpha$, $\tau$), and the effect of them is not considered or analyzed. There is only one table on $\alpha$. Particularly, I think the effect of $thr$ should be very important since the generative models are typically sensitive to the hyperparameter choice.
>
> >Exact details are not clearly presented and it is not clear how to reproduce the result.
> > - Line 176: how is thr chosen?
> > - Line 268: "remove some low-confidence samples" --> this is not rigorous.
>
> **A1**: Thanks for your concerns and suggestions, we added more analyses and details on the hyperparameters $thr$, $\alpha$, and $\tau$ (especially the hyperparameter $thr$) in Appendix B.
>
> - **Hyperparamter $thr$**
>
>   - **The effect of $thr$ on classifier and generator.** The pseudo-labels given by the classifier may be wrong and not all of the generated samples are high-quality. As a result, there may be some label noises. When we assign a pseudo-label to a sample, intuitively, a high Softmax score means less probability of being classified incorrectly. Therefore, to mitigate the label noises, we introduce the hyperparameter $thr$ to decide whether an image could be used as a training sample for classifier and generator.
>
>   - **The effect of $thr$ on representation model.** Before we perform fairness-aware contrastive learning, we first prepare contrastive sample pairs using the trained generator. Then we remove some low-confidence generated samples, since there are a few low-quality generated samples with wrong sensitive attributes, which may introduce bias in the training process of the representation model. Specifically, we compute the average Softmax score for each pair of contrastive samples, and the sample pairs with low average Softmax scores (less than $thr$) will be removed.
>
>    - **Ablation study on $thr$.** To explore the effect of hyperparameter $thr$, we did ablation studies by setting $thr$ to different values. As shown in the following table, we can observe that the model's Equal Odds (EO) gradually decreases with the increase of $thr$. This experimental results are consistent with our expectation, since large values of $thr$ guarantee the correctness of the sensitive attributes of the training samples. However, the increase of $thr$ results in a slight decrease in accuracy, which may be due to that the number and diversity of training samples is reduced.
>
>   - **How to choose $thr$?** There is no criterion to choose $thr$, since the datasets, generative models, training algorithms and requirements are not invariant. However, we can randomly sample from the data at different $thr$ to assess the corresponding quality of the data at different $thr$. In this paper, we set $thr=0.9$ to balance the fairness and accuracy. In practice, we recommend that the users who are very concerned about fairness set the $thr$ to a large value.
>
> |           Method          | _FairCL_ | _FairCL_ | _FairCL_ | _FairCL_ | _SimCLR_ |
> |:-------------------------:|:--------:|:--------:|:--------:|:--------:|:--------:|
> |           $thr$           |   0.99   |   0.90   |   0.80   |   0.00   |     /    |
> | Ratio of removed data (%) |   28.4   |   20.0   |   11.6   |    0.0   |    0.0   |
> |        Accuracy (%)       |   75.0   |   75.3   |   75.8   |   76.7   |   77.1   |
> |      Equal Odds (EO)      | **15.8** | **16.8** | **18.0** | **20.2** | **36.2** |
>
> - **Hyperparameter $\alpha$**
>
> We introduce the hyperparameter $\alpha$ to control the trade-off between accuracy and fairness for our proposed algorithms. We have did ablation study on $\alpha$ in Section 4.5. As shown in Table 3, a larger scaling parameter $\alpha$ can yield a fairer model but with lower accuracy.
>
> - **Hyperparameter $\tau$**
>
> Hyperparameter $\tau$ is a temperature parameter, which is widely used in contrastive loss computation for sharper distribution of the Softmax output. In this paper, we followed the setting in [1] and set $\tau=0.5$.
>
> **Corresponding Manuscript Revision:**  We have added more ablation studies on hyperparamters in Appendix B. Moreover, we have improved our presentation in Section 3.2 and Section 4.1.
>
> [1] Ting Chen, Simon Kornblith, Mohammad Norouzi, and Geoffrey Hinton. "A simple framework for contrastive learning of visual representations." ICML 2020.

---

> > ### Author Response · Authors · 2022-11-18
> > **Response to Reviewer kKgZ (2/3)**
> >
> > ## Q2: Generalization to non-facial dataset and dependencies on generative model.
> > > The method is highly dependent on the GAN-based generative model, and when the dataset is not face-related images like CelebA or UTKFace, it is not clear how the method would perform.
> >
> > **A2:** Thanks for your concerns. We added more experiments with different generative mothods, for example HFGI [3], in Appendix E. We also added extra expermiments on non-facial dataset in Appendix D. We hope the responses in the following can help to address your concerns.
> >
> > - **Extra experiments of generalization to the non-facial dataset.**
> >
> >   - **Experimental setup.** We used Dogs and Cats dataset [2], which is commonly used in the previous debiasing studies. We set color as the sensitive attribute, and the task is to predict if the image is a cat or a dog. We constructed a biased training set where the Pearson correlation coefficient is -0.33, so that the model may learn the spurious correlation between species and colors.
> >
> >   - **Experimental results.** The classification results are shown in the table as below. Our proposed *FairCL* improves fairness compared to *SimCLR* with only a slight drop in accuracy. Moreover, *FairCL* outperforms all baseline methods in terms of both fairness and accuracy. The above experimental results illustrate that our proposed framework can perform well on the non-facial dataset, and can prevent representation models from learning spurious correlations.
> >
> > | Dogs and Cats dataset, T=*species*, S=*color* |  EO  | Acc. (%) |
> > |:----------------------:|:----:|:--------:|
> > |       *CGL+VFAE*       | 10.3 |   85.9   |
> > |        *CGL+GRL*       |  9.7 |   86.6   |
> > |        *SimCLR*        | 12.4 |   88.1   |
> > |     *FairCL* (Ours)    |  **8.2** |   87.5   |
> >
> > - **Extra experiments of generalization to different generative methods.**
> >
> >   - **Experimental Setup.** We used another generative method, HFGI [3], to generate contrastive sample pairs. We used UTK-Face as the dataset and set race as the sensitive attribute. The task is to predict the gender. We constructed three unbalanced settings where the Pearson correlation coefficients of race and gender in the training dataset are -0.2, -0.4, and -0.6, respectively. We constructed a balanced (unbiased) test set.
> >
> >   - **Experimental Results.** We report the classification results in the table as below. We observe that our proposed _FairCL*_ improves *BYOL* in terms of both fairness and accuracy in three settings, which illustrates that the representation models trained by our framework is more effective and fairer. The results also show that our proposed fairness-aware framework is compatible with different generative methods. This means that we can choose the appropriate generation method according to the dataset and task.
> >
> >   - **Discussion.** The goal of our work is to learn good representations, while sample generation is just a tool to achieve this goal. It would be nice to be able to generate high-quality contrastive samples. However, it is possible to learn a good representation model even if the generated samples are not of very high quality. For example, as shown in Figure 8, the visual quality of the generated images is not as high as we would like, and the racial changes in the edited images are not very obvious. But the classification results of our method are still greatly improved compared with the baseline method. Moreover, we would like to emphasize that our main methodological contribution lies in proposing a new framework for fair contrastive learning rather than generative models. Note that the general framework is compatible with different generative methods, so that it will benefit from the development of generative techniques.
> >
> > |      Pearson correlation coefficient     |   -0.2  |   -0.2   |   -0.4  |   -0.4   |   -0.6  |   -0.6   |
> > |:----------------:|:-------:|:--------:|:-------:|:--------:|:-------:|:--------:|
> > |      UTK-Face dataset, T=*gender*, S=*race*      |    EO   | Acc. (%) |    EO   | Acc. (%) |    EO   | Acc. (%) |
> > |      *BYOL*      |   2.1   |   89.4   |   8.7   |   87.5   |   13.1  |   86.4   |
> > | _FairCL* (Ours)_ | **1.7** | **90.6** | **5.6** | **89.6** | **8.9** | **89.7** |
> > |    Improvement   |   -0.4  |    1.2   |   -3.1  |    2.1   |   -4.2  |    3.3   |
> >
> > **Corresponding Manuscript Revision:**  We have added additional experiments on a non-facial dataset (Dogs and Cats) in Appendix D. We also added experiments based on a new generative method (HFGI) in Appendix E.
> >
> > [2] Dogs vs. cats. In Kaggle, 2013.
> >
> > [3] Tengfei Wang, Yong Zhang, Yanbo Fan, Jue Wang, and Qifeng Chen. "High-fidelity gan inversion for image attribute editing."  CVPR 2022.

---

> > > ### Author Response · Authors · 2022-11-18
> > > **Response to Reviewer kKgZ (3/3)**
> > >
> > > ## Q3: Desription of the iterative algorithm and its effects.
> > > >In fact, the whole paragraph line 171~182 is not very rigorous. It is not clear how to reproduce based on the description. Also, it is not clear whether the iterative steps described in the paragraph would always work.
> > >
> > > **A3:** Thanks for your suggestion. We have improved the presentation (Lines 171~183). Moreover, we have added more detailed description of our method (the pseudo-code) in Appendix A to make the presentation clearer.
> > >
> > > Empirically, we also added ablation studies to show the effectiveness of the proposed algorithm (the iterative steps).  Firstly, we record the accuracy of the classifier with and without the above iterative steps. Compared with the situation without the above iterative steps, the accuracy of the classifier (used for sensitive attribute prediction) has been improved after the above iterative steps are introduced (from 87% to 97%, CelebA dataset, S=Young, $thr=0.9$). Secondly, we also find that the above iterative steps are helpful to improve the learned representation model in terms of fairness. Please refer to Appendix B, if you are intersted in more details.
> > >
> > > **Corresponding Manuscript Revision:** We have improved the presentation in Section 3.2 and Section 4.1. We also provided the detailed pseudo-code in Appendix A. Besides, we added ablation studies in Appendix B.
> > >
> > >
> > > ## Q4:  Novelty.
> > > >The problem setting seems to be new (but, it is also inspired by recent work (Jung et al, 2022) and is not very hard to think of). The components of the method are careful combination of existing techniques rather than novel development of new schemes. I think the overall method simply combines existing methods (which undermines the novelty).
> > >
> > > **A4:** Thanks for your concern. We would like to response the concern of novelty from the following aspects.
> > >
> > > - Firstly, the setting we propose is **new** and **meaningful**. The recent work [4] foucses on the setting where the target variable is supervised and the sensitive attribute is semi-supervised. Compared with it, our proposed setting is more **practical**, since the target labels are not required (unsupervised). Moreover, our setting is also more **challenging**, since there may be an unseen spurious correlation between unknown target variable and sensitive attributes.
> > >
> > > - Secondly, we propose a **novel framework (pipeline)** to solve the above problem. To the best of our knowledge, we are **the first one** to exploit generative techniques to construct positive/negative samples for self-supervised contrastive learning to learn fair representations. In additon, our framework is **general** and **compatible** with different contrastive learning methods and different generative methods, so that the the existing techniques can be used here.
> > >
> > > - Lastly, we also have several technical contributions on the submodules and implementations of the pipeline, such as: 1) proposing a semi-supervised learning algorithm for sample generation, and 2) provding a simple way to balance the fairness and accuracy in unsupervised contrastive learning.
> > >
> > > [4] Sangwon Jung, Sanghyuk Chun, and Taesup Moon. "Learning fair classifiers with partially annotated group labels." CVPR 2022.

---

### Author Response · Authors · 2022-11-18
**General Response to All Reviewers**

We thank all reviewers for recognizing our paper well-written (Reviewers MhQc, Vv8C, G4GB), easy to follow (Reviewers MhQc, Vv8C), with novel settings/ideas/methods (Reviewers kKgZ, MhQc, Vv8C), and with high-quality/enough experiments (Reviewers MhQc, Vv8C).

We appreciate their suggestions and comments, and the main concerns can be summaried as: 1) the presentation of exact details; 2) missing ablation studies on hyperparameters; 3) missing fairness-accuracy trade-off curve; 4) generzaiton to non-facial dataset; and 5) heavily relying on specific generative method. We have carefully revised our paper accordingly. Our major revisions are listed as follows:

+ For 1), we improved the presentation (Lines 171\~183, Lines 268\~270) and added the pseudo-code in Appendix A.
+ For 2), we added the ablation studies on hyperparameters in Appendix B.
+ For 3), we added the fairness-accuracy trade-off curves on CelebA dataset in Appendix C.
+ For 4), we added the experiments on a non-facial dataset in Appendix D.
+ For 5), we added the experiments based on a new generative method in Appendix E.

In addition, there are some minor changes:
+ We added a table to list the experiemental settings in Appendix F.
+ We added more discussion about the failure cases (Lines 287\~290).

Please note that we colorized the revisions in the new version of the paper.

---

### Author Response · Authors · 2022-12-03
**Thanks to Reviewers and AC**

Dear Reviewers and AC,

We sincerely appreciate your efforts in reviewing our paper, and your constructive comments. We have responded to your comments, faithfully reflected them in the revision, and provided additional experimental results that you have requested. We are wondering if there are any remaining questions or concerns we can address, and we are more than happy to answer them.

Best regards,

Paper276 Authors

---

### Decision · Program_Chairs · 2023-01-20

**Decision:**

Accept: poster

**Justification For Why Not Higher Score:**

The contributions of this paper appear sound and interesting. Although experimental validation is quite extensive and points towards more fair representations, it is limited to domains where we know generative models excel. It is unclear how the method would perform in less constrained domains, and in turn, in more realistic (and challenging) setups.

**Justification For Why Not Lower Score:**

The problem tackled in this paper is important (fair representation learning with partial annotations), the proposed approach is sound and the reported results rather compelling. The problem setting is novel and has practical value. The only remaining concern was in the generality of the proposed approach and how it would perform on more challenging datasets, where image generative models may not provide such high quality generations.

**Metareview: Summary, Strengths And Weaknesses:**

This submission was reviewed by four knowledgeable reviewers. The reviewers found the paper well written and easy to follow (MhQc, Vv8C, G4GB), the contributions sound (kKgZ, MhQc, Vv8C, G4GB) and rather novel (kKgZ, MhQc, Vv8C). The reviewers raised concerns wrt the dependence of the method on high quality image generations (kKgZ, Vv8C, G4GB), the unclear influence of the many hyper-parameters (kKgZ), the potential biases introduced by training the image editor and the sensitive label classifier with unlabeled data (MhQc), the seemingly limited application scope (G4GB), and the fairness-accuracy trade-offs (G4GB). The authors proactively engaged with the reviewers and addressed most of their concerns. During the discussion phase, reviewers appreciated the additional experiments and the efforts made by the authors (e.g. ablations, new generation model, experiments beyond faces) to improve their submission. The only remaining concern is the scope of the presented experiments, which focus on faces and which have been complemented with cats and dogs during discussion. The effectiveness of the proposed approach on more challenging and unconstrained domains remains to be demonstrated, in part due to the potentially lower-quality generations in more diverse domains. Despite these limitations, there is a general agreement that the contribution is worthy of acceptance. The AC agrees with the reviewers and recommends to accept. However, as per discussion with the reviewers, the AC strongly recommends to include a limitation section, including a detailed discussion on the limitations mentioned by the reviewers and the future works to address those.

**Note From Pc:**

if the above contains the word "oral" or "spotlight" please see: "oral" presentation means -> notable-top-5% and "spotlight" means -> notable-top-25%. As stated in our emails, we are disassociating presentation type from AC recommendations